# MASK-TUNING: TOWARDS IMPROVING PRE-TRAINED LANGUAGE MODELS' GENERALIZATION

## ABSTRACT

Pre-trained language models have the known generalization problem. This issue emerges from the pre-trained language models' learning process that heavily relies on spurious correlations, which work for the majority of training examples but do not hold in general. As a consequence, the models' performance drops substantially on out-of-distribution datasets. Previous studies proposed various solutions, including data augmentation and learning process improvement. In this paper, we present **Mask-tuning**, an approach that alleviates the impact of spurious correlations on the fine-tuning learning process. To achieve this goal, Mask-tuning integrates masked language training into the fine-tuning learning process. In this case, Mask-tuning perturbs the linguistic relation of downstream tasks' training examples and computes masked language training loss. Then, the perturbed examples are fed into fine-tuning process to be classified based on their ground-truth label and compute the fine-tuning training loss. Afterward, Mask-tuning loss– a weighted aggregation of masked language model training loss and fine-tuning loss– updates the masked language model and fine-tuning through training iterations. Extensive experiments show that Mask-tuning consistently improves the pre-trained language models' generalization on out-of-distribution datasets and enhances their performance on in-distribution datasets. The source code and pre-trained models will be available on the author's GitHub page.

## 1 INTRODUCTION

One of the challenges in building a pre-trained language model with robust generalization is that training sets do not represent the linguistic diversity of real-world language. Thus its performance dramatically drops when encountering out-of-distribution datasets. This type of performance divergence on the in-distribution to out-of-distribution datasets is named *generalization gap*. Previous studies (Zhang et al., 2019; McCoy et al., 2019; Tu et al., 2020) have shown that pre-trained language models trained on the specific dataset are likely to learn spurious correlations, which are prediction rules that work for the majority examples but do not hold in general. It means that the fine-tuning loss function cannot incentivize the language model to learn the linguistics patterns from the minority examples and generalize them to more challenging examples (e.g., Out-of-distribution dataset). There are several solutions that have been referred the generalization gap as: annotation artifacts (Gururangan et al., 2018), dataset bias (He et al., 2019; Clark et al., 2019; Mahabadi et al., 2020), spurious correlation (Tu et al., 2020; Kaushik et al., 2020), and group shift (Oren et al., 2019). However, most of these methods rely on the strong assumption of knowing the datasets' spurious correlations or biased keywords in advance and also suffer from decreasing the model performance on the in-distribution dataset.

One of the primary solutions for mitigating the generalization gap is directly increasing the number of minority examples in the training set and creating a more balanced training dataset, which has been performed in several studies named: data augmentation (Ng et al., 2020; Garg & Ramakrishnan, 2020), domain adaptation

(Gururangan et al., 2020; Lee et al., 2020), multi-task learning (Pruksachatkun et al., 2020; Phang et al., 2020), and adversarial data generation (Garg & Ramakrishnan, 2020; Li et al., 2021). Although these solutions have achieved different levels of success, they suffer from the following challenges: a) They require coming up with large, related, and useful training examples. b) They assume that an appropriate intermediate task and dataset are readily available. c) They need to re-run the pre-training phase from scratch. Finally, d) some of these methods added a complex procedure to the learning process. All these requirements need significantly high computational resources or human annotations (Zhou & Bansal, 2020; Chen et al., 2021).

In this paper, we propose *Mask-tuning* which is inspired by recent studies in computer vision (Hendrycks et al., 2020b; Mohseni et al., 2020) that showed joint-training improves the robustness of deep learning models. However, computer vision studies used a mix of in-distribution and out-of-distribution training examples for the training processes (labeled and unlabeled) besides changing the model's architecture by adding multiple auxiliary heads. In contrast, our approach (Mask-tuning) employs the original pre-trained language models' masked language modeling and fine-tuning and solely uses the downstream task's training dataset as labeled and unlabeled datasets. As a result, Mask-tuning is applicable with any pre-trained language model that works with the original fine-tuning.

Mask-tuning reinforces the learning process of the pre-trained language models' fine-tuning by integrating the masked language model training into the fine-tuning process. The proposed approach hinders learning the spurious correlations by using masked language model loss to incentivize the fine-tuning training loss when the correct output likely be according to the learned spurious correlations. We used masked language modeling to perturb the linguistics relation between input training examples by masking a certain percentage of input tokens, predicting those masked ones, and computing the masked language model loss. Then the perturbed example is fed to fine-tuning for classification according to the ground-truth label. Afterward, Mask-tuning loss– a weighted aggregation of masked language modeling loss and fine-tuning loss– updates the masked language model and fine-tuning through training iterations. Mask-tuning–especially the fine-tuning step after masked language modeling–learns that training examples with different linguistic relations can have the same semantics and belong to the same label. Our main contributions are as follows:

- We study the effect of integrating the masked language modeling into fine-tuning training process on the pre-trained language models' generalization and proposed Mask-tuning. Our proposed method is a plug-and-play tool applicable to any pre-trained language model without the need for making any changes to the pre-trained language models' architecture. Indeed, our approach solely uses the downstream task's training dataset. To our knowledge, we are the first to integrate the masked language model into the fine-tuning learning process that includes training losses aggregation.

- Conducting comprehensive experiments under a consistent evaluation process to verify the effectiveness of Mask-tuning. Using BERT, RoBERTa, and autoregressive BART language models, we show that Mask-tuning outperforms eight state-of-the-art approaches from literature on three downstream tasks and three in-distribution and five out-of-distribution datasets.

- Our ablation studies show the necessity of simultaneously running masked language model training with fine-tuning in mask-tuning. Extensive experiments show that when Mask-tuning trains only by fine-tuning loss or applying each training step separately (only fine-tuning the perturbed training examples), it could not mitigate the generalization gap as well as Mask-tuning with joint-loss.

## 2   APPROACH

In this section, we formally introduce the proposed approach setup named Mask-tuning. We first define the details of the Mask-tuning setup that is also illustrated in Fig.1. Then we explain the perturbation strategy and insights behind the Mask-tuning.

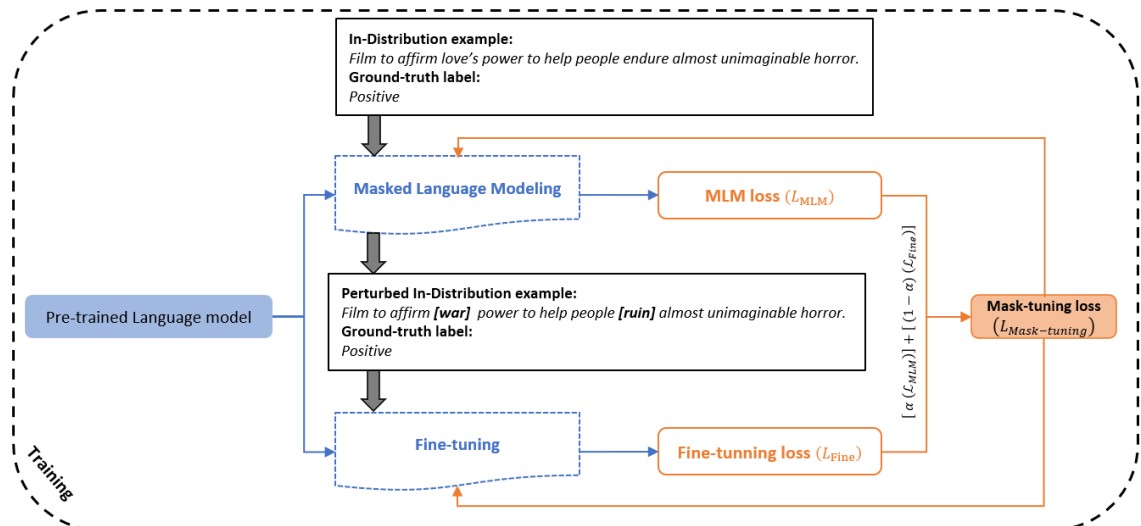

Figure 1: Illustration of the Mask-tuning's training process, a combination of the pre-trained language model's Masked language modeling and Fine-tuning. The input of the fine-tuning is a perturbed version of the in-distribution training example. Both Masked language modeling and Fine-tuning are trained based on the Mask-tuning loss.

## 2.1 MASK-TUNING

As shown in Figure1, Mask-tuning combines two training steps, masked language model and fine-tuning, to enhance the training process of the fine-tuning to learn more linguistics patterns besides spurious correlations. For this aim, Mask-tuning perturbs the relation between the input tokens using the masked language model and trains the fine-tuning to classify the perturbed example based on the ground-truth label through the following steps:

First, Mask-tuning initiates with masking a certain percentage of the input tokens at random and then predicts those masked ones. In this case, the final hidden vectors corresponding to the mask token(s) is fed into an output softmax over the embedding vocabulary as in a standard language model. In all of our experiments, we randomly mask 5% of all tokens in each input sequence. If the $i$-th token(s) is chosen, Mask-tuning replaces the $i$-th token(s) with the [MASK] token(s). Then the final hidden vector for $i$-th token(s) will be used to predict the masked token(s) with the aggregation of the cross-entropy loss from all masked token(s) that we denote as Masked Language Model training loss ($\mathcal{L}_{MLM}$)(Fig. 1) .

Second, the training example with the predicted token(s), called *perturbed example*, fed into fine-tuning to be classified based on the ground-truth label ($y$). Then $p_\theta(y' = y|\hat{x})$ is the fine-tuning function to predict the perturbed example's label ($y'$) based on the perturbed example ($\hat{x}$) and compute the fine-tuning training loss ($\mathcal{L}_{Fine}$), where $\theta$ is the pre-trained language model parameters for the fine-tuning. Afterward, a weighted aggregation of these two processes (i.e., masked modeling and fine-tuning) computes Mask-tuning loss ($\mathcal{L}_{Mask-tuning}$) as follows:

$$\mathcal{L}_{Mask-tuning} = \alpha\,\mathcal{L}_{MLM} +\ (1-\alpha)\mathcal{L}_{Fine} \qquad (1)$$

Where $\alpha$ is a weighting factor, we employ it to adjust the contribution of the two training losses in computing

the Mask-tuning loss. For passing each training iteration, the training loss of both steps must be close to zero. Otherwise, the training continues to minimize Mask-tuning training loss. Thus, on the one hand, Mask-tuning loss forces masked language modeling to predict a token semantically close to the original token, but with linguistic variations that still allows fine-tuning to succeed. On the other hand, Mask-tuning avoids fine-tuning to be rewarded for being correct for the completed wrong predicted tokens (Section 2.3).

## 2.2 PERTURBATION STRATEGY

There are different noise injection strategies, such as additive Gaussian noise, dropout noise, or simple data augmentation; each matches a particular downstream task. In this study, we follow some data-augmentation studies (e.g., Garg & Ramakrishnan, 2020) that used masked-language modeling to directly increase the size of the training datasets and train the model on a more variety of examples. In contrast, we employ masked language modeling as a robust noise injection method used in unsupervised pre-training in the pre-trained language models such as BERT. We hypothesize that the successful masked language modeling that achieves state-of-art performance on in-distribution datasets generates realistic and diverse perturbed examples that considerably modify the relation between the input token without affecting the label. Thus, it is safe to induce consistency between two training phases,i.e., masked language modeling and fine-tuning. Indeed, the extreme experiment results show that Mask-tuning's perturbation strategy matches different downstream tasks.

## 2.3 THE EFFECT OF MASK-TUNING ON THE LEARNING PATTERNS

The key intuition behind Mask-tuning is breaking the linguistic relation between the words in an example and classifying the perturbed examples according to the ground-truth label. Then based on the perturbed examples, Mask-tuning learns that training examples with different linguistic relations can have the same semantics (label). Thus it makes the model robust when applied to out-of-distribution (unseen) examples. For passing training iteration, both training steps' losses ( i.e., 1) masked language and 2) fine-tuning) must be close to zero (output is correct), and Mask-tuning loss is close to zero. Otherwise, suppose the training loss of masked modeling is large and fine-tuning classifies the perturbed example correctly (fine-loss close to zero), likely based on spurious correlations or statistical patterns. In this case, masked language modeling loss incentivizes the fine-tuning loss to be updated even with the correct output. We illustrate the effect of Mask-tuning loss by looking at the following scenarios:

▷ **The masked language modeling training loss is close to zero** ($\mathcal{L}_{MLM} \approx \mathbf{0}$ ); hence masked modeling is predicting the original token or a token that may be a close synonym. If the fine-tuning classifies the example correctly ($\mathcal{L}_{Fine} \approx 0$), the Mask-tuning loss ($\mathcal{L}_{Mask-tuning}$) is also close to zero, and there is no need for further training. Otherwise, if the output of any of these steps is not correct, Mask-tuning loss is large enough to prompt masked modeling to generate further variations and fine-tuning model to train further to handle variations. Thus it makes it harder for fine-tuning to rely only on the majority examples of the given downstream in-distribution dataset.

▷ **The masked language modeling training loss is significant** ($\mathcal{L}_{MLM} > \mathbf{0}$ ); suppose, the predicated token(s) is not semantically related to the original token(s) and raise masked language modeling loss (MLM-loss). In this case, the Mask-tuning loss is large enough, regardless of the fine-tuning classification output, to force the Mask-tuning training's models to continue training to minimize the Mask-tuning loss.

## 3 EXPERIMENT

### 3.1 IN-DISTRIBUTION AND OUT-OF-DISTRIBUTION TASKS AND DATASETS

In a typical fine-tuning setting, the language model is fine-tuned on the training examples from the downstream task and evaluated on a test/dev set drawn from the same distribution as the training set. However, for this study, we evaluated the fine-tuned model on the Dev set of the out-of-distribution datasets as well as in-distribution Dev sets. Out-of-distribution datasets are drawn from a different distribution with more linguistic variation than the in-distribution training set. Pre-trained language models' performance on out-of-distribution datasets indicates their robustness and generalizability.

In this study, experiments have been conducted on three tasks from GLUE (Wang et al., 2019) paraphrase Identification (PI), Natural Language Inference (NLI), and Stanford Sentiment Treebank (SST-2). They have large-scale benchmark datasets, and their corresponding out-of-distribution datasets are publicly available.
**Sentiment**. Stanford Sentiment Treebank (SST-2) has been selected as an in-distribution dataset for the sentiment classification task. For out-of-distribution counterparts, the recent challenging sentiment classification datasets of IMDB Contrast set (IMDB-Cont)[1] (Gardner et al., 2020) and IMDB Counter-factually Augmented Dataset (IMDB-CAD)[2] (Kaushik et al., 2020) have been selected.
**Natural Language Inference (NLI)**. Multi-Genre Natural Language Inference (MNLI)(Williams et al., 2018) is a popular NLI task's dataset containing about 400k premise/hypothesis pairs annotated with textual entailment information (neutral, entailment, and contradiction). MNLI includes two different dev sets, *matched* set (MNLI-m), derived from the same sources as those in the training set and *mismatched* set (MNLI-mm) that do not closely resemble any of those seen at training time (Williams et al., 2018). pre-trained models were trained on MNLI as an in-distribution dataset and then evaluated on MNLI dev sets (m/mm) and two relevant out-of-distribution datasets, HANS[3] and Adversarial NLI (AdvNLI)[4] .
**Paraphrase Identification (PI)**. Quora Question Pairs (QQP)(Iyer et al., 2017) is a widely used PI task's dataset containing around 400k pairs of sentences annotated as either paraphrase or non-paraphrase. Mask-tuning is trained on QQP as a PI task's in-distribution dataset and evaluated on QQP dev set and relevant out-of-distribution PAWS-QQP[5] (Zhang et al., 2019) with high lexical overlap but different semantic meanings.

### 3.2 EXPERIMENT SETUP

Three pre-trained language models have been chosen for this study, BERT (Devlin et al., 2019), RoBERTa (Liu et al., 2019b), and BART (Lewis et al., 2020). Note that BART is an autoregressive language model with a different architecture than the other two. Recent studies (e.g., Hendrycks et al., 2020a) illustrated that although language models of large sizes often perform well on both the in-distribution and out-of-distribution datasets, in comparison with base size language model, the larger model does not necessarily reduce the generalization gap. Thus we choose the base size of these models to show the efficacy of the Mask-tuning method. Besides, the base size models are more efficient with a smaller size, making them deployable to a broader computational environment and applications.

We follow the original language model's parameters [6] for two training steps of Mask-tuning. Except for fine-tuning step, learning-rate and batch size are changed based on our experiment. After trying 8,16, and 32

---

[1]https://github.com/allenai/contrast-sets/tree/main/IMDb

[2]https://github.com/acmi-lab/counterfactually-augmented-data

[3]https://github.com/tommccoy1/hans

[4]https://github.com/facebookresearch/anli

[5]https://github.com/google-research-datasets/paws

[6]https://github.com/huggingface/transformers

Table 1: Comparing the generalization performance on the out-of-distribution datasets using Mask-tuning (our study) and some previous approaches on three different language models, BERT, RoBERTa, and BART. All models are trained on related in-distribution dataset and then evaluated on dev set of out-of-distribution examples. The column "Type" shows that each method is related to which categories in the litrature (Section 6).The Fine-tuning (Original model) has been implemented using https://github.com/huggingface/transformers.

| | | | Out-Of-Distribution | | | | |
| | | | Sentiment | | NLI | | Paraphrase |
| Model | Method | Type | IMDB-Cont. | IMDB-CAD | HANS | AdvNLI | PAWS |
|---|---|---|---|---|---|---|---|
| BERT$_{base}$ | Fine-tuning | original | 79.08 | 87.00 | 56.90 | 24.12 | 32.80 |
| | Learned-Mixin+H (Clark et al., 2019) | Joint-train | - | - | 66.15 | - | - |
| | DRiFt-Hand (He et al., 2019) | Debiasing | - | - | 66.20 | - | - |
| | PoE (Mahabadi et al., 2020) | Joint-train | - | - | 66.31 | - | - |
| | Regularized-conf (Utama et al., 2020) | Debiasing | - | - | 69.10 | - | 39.80 |
| | IPT-standard (Glavaš & Vulić, 2021) | Data-aug. | - | - | 56.70 | - | - |
| | Mask-tuning | Our study | 82.75±0.3 | 88.32±0.1 | 69.52±0.2 | 26.32±0.3 | 46.74±0.5 |
| RoBERTa$_{base}$ | Fine-tuning | original | 84.50 | 88.40 | 67.80 | 31.20 | 38.45 |
| | Span Cutoff (Shen et al., 2020) | Data-aug. | 85.50 | 89.20 | 68.40 | 31.10 | 38.80 |
| | HiddenCut (Chen et al., 2021) | Data-aug. | 87.80 | 90.40 | 71.20 | 32.80 | 41.50 |
| | IPT-Adapter (Glavaš & Vulić, 2021) | Data-aug. | - | - | 66.30 | - | - |
| | Mask-Tuning | Our study | 88.50±0.2 | 91.62±0.1 | 75.70±0.2 | 37.40±0.6 | 44.37±0.4 |
| BART$_{base}$ | Fine-tuning | Original | 82.48 | 86.03 | 56.30 | 30.51 | 32.27 |
| | Mask-Tuning | Our study | 83.00±0.1 | 87.83±0.3 | 70.48±0.5 | 35.31±0.4 | 45.71±0.5 |

for batch size in several trials run, the batch size of 32 is used for double sentence datasets (i.e., MNLI and QQP) and 16 for single sentence datasets (i.e., SST-2). The best learning rate has been determined through grid search among $\{2 \times 10^{-5}, 3 \times 10^{-5}, 4 \times 10^{-5}, 5 \times 10^{-5}\}$. We empirically selected the optimal value for $\alpha$ by a grid search in $0 < \alpha < 1$ with 0.1 increments. For each downstream task, the best value of $\alpha$ is o.6, 0.7, and 0.8 for SST, NLI, and PI, respectively. Also, through grid search, the masking percentage is set to 5%, which achieved the best performance on in-distribution datasets. All experiments were performed with 3 epochs and using an NVIDIA V100 GPU with five random seeds, and the average values for the five experiments with corresponding standard deviations are reported. We used Wolf et al. (2020)[7] for the original Fine-tuning.

# 4 RESULTS

In each evaluation setup, we have reported the results of prior studies on in-distribution and out-of-distribution datasets for comparison purposes. According to the literature, these previously proposed methods belong to two categories: data augmentation and learning process improvement. According to the literature (Section 6.3), the previous studies on learning process improvement introduce a new learning method through joint training or debiasing algorithm. As it can be seen from the Tables 1 and 2, only Glavaš & Vulić (2021) reported their results on both BERT and RoBERTa models, and the others applied just one of them. Indeed we report the accuracy results of the Mask-tuning when using the BART language model with a different architecture than BERT on in-distribution and out-of-distribution datasets. To the best of our knowledge, no study has reported BART model performance on out-of-distribution datasets. Thus we compare our proposed method's result with the original fine-tuning performance on out-of-distribution data.

## 4.1 PERFORMANCE ON OUT-OF-DISTRIBUTION DATASETS

The evaluation accuracy results on out-of-distribution datasets have been shown in Table 1. Mask-tuning gains the state-of-the-art performance on all three pre-trained language models, BERT, RoBERTa, and BART on all tasks and out-of-distribution datasets. To the best of our knowledge, no study has reported BERT-base

---

[7]https://github.com/huggingface/transformers

Table 2: The performance on the in-distribution datasets using Mask-tuning (our study) and some previous approaches on three different language models, BERT, RoBERTa, and BART. The results show Mask-tuning did NOT hurt the models' performance on the in-distribution datasets. The Fine-tuning (original model) has been implemented using https://github.com/huggingface/transformers.

| | | In-Distribution | | |
| | | Sentiment | NLI | Paraphrase |
| **Model** | **Approach** | **SST-2** | **MNLI** (m/mm) | **QQP** |
| $\text{BERT}_{base}$ | Fine-tuning (Original model) | 92.43 | 84.30/83.40 | 90.80 |
| | Learned-Mixin+H (Clark et al., 2019) | - | 83.97/- | - |
| | DRiFt-Hand (He et al., 2019) | - | 81.70/- | - |
| | PoE (Mahabadi et al., 2020) | - | 84.19/- | - |
| | Regularized-conf (Utama et al., 2020) | - | 84.3/84.8 | 91.50 |
| | IPT-standard (Glavaš & Vulić, 2021) | - | 84.40/- | - |
| | Mask-tuning (Our study) | **93.11**±0.1 | **84.75**±0.2/**85.10**±0.1 | **91.54**±0.1 |
| $\text{RoBERTa}_{base}$ | Fine-tuning (Original model) | 94.49 | 87.60 / 87.50 | 91.50 |
| | Span Cutoff (Shen et al., 2020) | 95.40 | 88.40/- | 92.00 |
| | HiddenCut (Chen et al., 2021) | 95.80 | 88.20/- | 92.00 |
| | IPT-Standard ((Glavaš & Vulić, 2021) | - | 87.70/- | - |
| | Mask-tuning (Our study) | **94.60**±0.1 | **87.72**±0.1 / **87.83**±0.2 | **91.62**±0.2 |
| $\text{BART}_{base}$ | Fine-tuning (Original model) | 93.23 | 84.60/84.80 | 90.50 |
| | Mask-tuning (Our study) | **93.80**±0.2 | **86.08**±0.2 /**86.12** ±0.3 | **91.03**±0.1 |

model performance on out-of-distribution datasets of the sentiment classification task, IMDB-Cont, and IMDB-CAD. Thus, we compare the original BERT fine-tuning and our approach.

In comparison with the BERT original fine-tuning, Mask-tuning improves the performance on IMDB-Cont by +3.67, IMDB-CAD by +1.32, HANS by +12.62, AdvNLI by +2.2, and PAWS by +13.94. Moreover, Mask-tuning mitigates the generalization gap in RoBERTa model by improving original fine-tuning on IMDB-Cont by +4, IMDB-CAD by +3.22, HANS by +7.9, AdvNLI by +6.2, and PAWS by +5.92. And on BART original fine-tuning, Mask-tuning enhances the performance on IMDB-Cont by +0.52, IMDB-CAD by +1.8, HANS by +14.18, AdvNLI by +4.8, and PAWS by +13.44.

### 4.2 PERFORMANCE ON IN-DISTRIBUTION DATASET

Table 2 shows the performance of the different approaches on the in-distribution datasets in three pre-trained language models BERT, RoBERTa, and BART. It is an essential factor for evaluating an approach not to drop the performance of a pre-trained language model on in-distribution datasets while improving its performance on the out-of-distribution datasets. As it can be seen from Table 2, in contrast with some of the previous approaches, Mask-tuning gains state-of-the-art on the out-of-distribution datasets without hurting the performance on the in-distribution datasets but even boosting it.

## 5 ABLATION STUDY

We perform ablation experiments to demonstrate the necessity of each component of the Mask-tuning for improving the pre-trained language models' generalization. The proposed method added two critical parts, masked language modeling and Mask-tuning loss, to the fine-tuning learning process to alleviate the generalization problem. We perform two experiments to show the impact of the two components on the achieved performance. Two experiments are as follows:
▷ **Mask-tuning with Fine-tuning Loss** ($\mathcal{L}_{Fine}$) . In this experimental setup, we investigate the effect of masked language modeling loss ($\mathcal{L}_{MLM}$) on the performance of the proposed method. We ran Mask-tuning

Table 3: Ablation results over the effectiveness of the joint-training loss in Mask-tuning. Mask-tuning with ($\mathcal{L}_{Fine}$) means the language model only trained with the fine-tuning loss. And Mask-tuning without Mask-tuning loss ($\mathcal{L}_{Mask-tuning}$) means we first generate the perturbed examples and then fine-tune the language model based on the perturbed examples.

| | Sentiment | | | NLI | | | Paraphrase | |
|---|---|---|---|---|---|---|---|---|
| | In-Dis. | Out-of-Dis. | | In-Dis. | Out-of-Dis. | | In-Dis. | Out-of-Dis. |
| **Model** | SST-2 | IMDB-Cont. | IMDB-CAD | **MNLI (m/mm)** | **HANS** | **AdvNLI** | **QQP** | **PAWS** |
| **BERT**$_{base}$ | | | | | | | | |
| Fine-tuning (Original model) | 92.43 | 79.08 | 87.00 | 84.30/83.40 | 56.90 | 24.12 | 90.80 | 32.80 |
| Mask-tuning w $\mathcal{L}_{Fine}$ | 92.07 | 79.00 | 84.99 | 83.57/84.22 | 50.51 | 23.60 | 89.92 | 38.15 |
| Mask-tuning w/o $\mathcal{L}_{Mask-tuning}$ | 91.97 | 80.97 | 86.76 | 83.40/84.00 | 55.60 | 24.20 | 91.00 | 34.82 |
| Mask-tuning | **93.11**±0.1 | **82.75**±0.3 | **88.32**±0.1 | **84.75**±0.2/**85.10**±0.1 | **69.52**±0.2 | **26.32**±**0.3** | **91.54**±0.1 | **46.74**±0.5 |
| **RoBERTa**$_{base}$ | | | | | | | | |
| Fine-tuning (Original model) | 94.49 | 84.50 | 88.40 | 87.60/87.50 | 67.80 | 31.20 | 91.50 | 38.45 |
| Mask-tuning w $\mathcal{L}_{Fine}$ | 93.00 | 81.80 | 87.31 | 86.88/87.24 | 65.16 | 30.00 | 90.09 | 41.00 |
| Mask-tuning w/o $\mathcal{L}_{Mask-tuning}$ | 93.26 | 84.86 | 89.00 | 87.50/87.40 | 71.69 | 30.70 | 91.20 | 39.91 |
| Mask-tuning | **94.60**±0.1 | **88.50**±0.2 | **91.62**±0.1 | **87.72**±0.1/**87.83**±0.2 | **75.70**±0.2 | **37.40**±0.6 | **91.62**±0.2 | **44.37**±0.4 |
| **BART**$_{base}$ | | | | | | | | |
| Fine-tuning (Original model) | 93.23 | 82.48 | 86.03 | 84.60/84.80 | 56.30 | 30.51 | 90.50 | 32.27 |
| Mask-tuning w $\mathcal{L}_{Fine}$ | 93.00 | 82.02 | 85.80 | 84.02/84.10 | 55.12 | 32.40 | 89.34 | 35.21 |
| Mask-tuning w/o $\mathcal{L}_{Mask-tuning}$ | 92.88 | 82.65 | 86.13 | 84.81/84.25 | 58.00 | 32.50 | 89.95 | 34.62 |
| Mask-tuning | **93.80**±0.1 | **83.00**±0.1 | **87.83**±0.3 | **86.08**±0.2/**86.12**±0.3 | **70.48**±0.5 | **35.31**±0.4 | **91.03**±0.1 | **45.71**±0.5 |

on in-distribution downstream dataset and set the $\alpha$ to 0 (Equation 1). Thus, the Mask-tuning training model is updated only based on fine-tuning loss through this experiment. As we can see from Table 3, eliminating masked language modeling loss from the mask-tuning loss hurts the Mask-tuning performance on both in-distribution and out-of-distribution datasets. Also the results show that the Mask-tuning with fine-loss could not improve the original fine-tuning performance but even decrease it. Hence, using masked modeling loss in the Mask-tuning learning process is essential. Furthermore, this experiment implies the impact of the mask-tuning loss to hinder the fine-tuning to predict only based on learned spurious correlation.

▷ **Mask-tuning without Mask-tuning Loss ($\mathcal{L}_{Mask-tuning}$).** This experiment aims to show the significance of running three components of Mask-tuning ( i.e., input perturbation, fine-tuning, and computing Mask-tuning loss) simultaneously to improve the generalization of the fine-tuning's learning process. We initialize the experiment by applying masked language modeling and generating the perturbed examples. Then run the fine-tuning on perturbed examples. As we can see from Table 3, the performance of the Mask-tuning without Mask-tuning loss dropped dramatically. However, because of two rounds of separate training (first masked modeling and second fine-tuning) on the in-distribution dataset, this setting could slightly improve the original fine-tuning performance in some datasets.

the performance of the Mask-tuning without Mask-tuning loss dropped dramatically. However, because of two round of separate training (first masked modeling and second fine-tuning) on the in-distribution dataset, this setting could slightly improve the original fine-tuning performance in some datasets.

# 6 PREVIOUS WORK

## 6.1 TRANSFER LEARNING

Recently, transfer learning has become a popular solution to build a more robust unsupervised pre-training (e.g., BERT (Devlin et al., 2019), RoBERTa (Liu et al., 2019b)) for processing the challenging examples. Transfer learning for improving the pre-trained language models' generalization has been implemented through two different approaches, domain- and task-adaptation. The domain-adaptation continues pre-training on examples with the same domain as the downstream task dataset. Some studies proposed to pre-train the language models in their domain of interest from scratch (e.g., (Huang et al., 2019; Lee et al., 2020)). In contrast, other studies suggested adding data from the domain of interest to the existing pre-training data (e.g.,

(Alsentzer et al., 2019; Gururangan et al., 2020)). Besides, task-adaptation focuses on sharing knowledge across different tasks (Liu et al., 2019a; Pruksachatkun et al., 2020; Vu et al., 2020). In task-adaptation, the pre-trained model first fine-tunes on a dataset most related to the target task and then fine-tuned again on the target task. The two studies have shown the benefits of transfer learning approaches, especially when target training data is limited. However, pre-training on a large-scale dataset is computationally expensive. Furthermore, the transfer learning approach's effectiveness is highly limited to the size of the data, the similarity between the source and target tasks and domains, and task complexity (Vu et al., 2020).

## 6.2 DATA AUGMENTATION

The most straightforward approach to improve the pre-trained language models' robustness is data augmentation. In this method, the downstream task dataset is directly enriched with examples from the target distribution using various techniques such as: increasing the size of the training dataset (Ng et al., 2020; Khashabi et al., 2020), balancing the existing cues (Schuster et al., 2019), adversarial training (Jia & Liang, 2017; McCoy et al., 2019; Garg & Ramakrishnan, 2020; Li et al., 2021), augmenting the standard training dataset with syntactic information (Min et al., 2020; Glavaš & Vulić, 2021), creating partial view to augment the training data (Shen et al., 2020), or dropping span of hidden information(Chen et al., 2021). Despite pre-trained language models benefit from data augmentation, collecting new datasets, especially at a large scale, is costly. Also, the size of the required data for the best performance is fuzzy. Unlike traditional data augmentation, our Mask-tuning does not use additional training examples. Instead, it perturbs the original examples while keeping the size of the training dataset constant.

## 6.3 LEARNING PROCESS IMPROVEMENT

Another track of study to enhance the pre-trained models' generalization is introducing new learning techniques that are robust to cope with dataset bias (Belinkov et al., 2019; He et al., 2019; Clark et al., 2019; Oren et al., 2019; Schuster et al., 2019; Mahabadi et al., 2020; Zhou & Bansal, 2020; Moon et al., 2021). These techniques focus on challenging examples or keywords that do not allow the pre-trained models to make shortcuts during training. Thus they need to design processes to recognize and handle the biased patterns and keywords in the training examples and re-training the model on extra text corpora. These changes often make the model more complicated and computationally expensive. However, the main weakness of these methods is their strong assumption on knowing the datasets biases or biased keywords in advance (Tu et al., 2020).

## 7 CONCLUSION

This study proposed Mask-tuning, a simple but effective approach for improving pre-trained language models' generalization. Mask-tuning employs masked language modeling for firstly perturbing the training example. As a result, this perturbation interrupts the linguistic relation between the input tokens. Secondly, the error of the perturbation process will play an inductive role in fine-tuning training. Mask-tuning learns to classify the perturbed examples according to their ground-truth label. Finally, Mask-tuning trains the language model to be robust when encountering a change in the input training example. Also, Mask-tuning has overcome the limitation of the previous methods. The comprehensive experiments on three in-distribution datasets and five out-of-distribution datasets from three different downstream tasks on three pre-trained language models (i.e., BERT, RoBERTa, and BART) illustrated that Mask-tuning outperformed the performance of the previous techniques on out-of-distribution datasets. The results also indicated that this generalization improvement was achieved without hurting their performance on in-distribution datasets but even boosting it.

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
