# OpenReview forum: "Mask-tuning: Towards  Improving  Pre-trained Language Models' Generalization"
_ICLR.cc/2023/Conference — Submitted to ICLR 2023_

### Official Review · Reviewer_Kw5T · 2022-10-24

**Confidence:** 5
**Correctness:** 2
**Technical Novelty And Significance:** 2
**Empirical Novelty And Significance:** 2
**Recommendation:** 3

**Clarity, Quality, Novelty And Reproducibility:**

The paper is very poorly written, and it is clear it is rushed towards the deadline, as it requires a substantial amount of rewrite to achieve the amount of clarity required from a conference paper.

The paper is riddled with typos and grammaticality issues.

## Section 2.3

- Improper usage of the term "semantics" -> sentences with same label doesn't necessarily mean they have the same "meaning".
- "passing training iteration" -> what does this line even mean?
- "fine-loss" -> fine-tuning loss
- "may be close to a synonym" -> why an assumption? This is related to the clarity issue of the perturbation strategy I highlighted above.
- Multiple grammatical issues in the whole section

## Section 3.2

- "base size models are more efficient with a smaller size" -> What does the author mean here? How does smaller models become more efficient?
- Typo in "For each downstream task, the best value of alpha is o.6" -> "0.6"

## Section 5

- This is probably the biggest proof that this paper is rushed: the entirety of the last three lines of the section are duplicated!



**Strength And Weaknesses:**

## The good

The fine-tuning technique makes sense on why it should improve the performance. The authors also include thorough experiments on an array of in-distribution and out-of-distribution datasets, which is commendable. The experimental results are also sound and contains the average and standard deviation over multiple runs, hence exhibiting robustness. Some models do get a substantial amount of improvement using their method, compared to the baseline.

## The bad

- Methodologically, the approach is similar to Domain/Task Adaptive Pre-training, where better downstream performance is achieved by continuing pre-training on the fine-tuning corpus (cite:gururangan2020). Hence, it is not suprising that this approach yields better results. The authors cite this work in a different context, but fail to draw similarities/dissimilarities with it.
- Different from TAPT/DAPT, authors here introduce "perturbation" on the input. It is not clear what is the motivation behind it. The authors directly jump into introducing a "perturbed" example in Section 2.1, without motivating the reader why it is needed.
- I'm not clear with what exactly the perturbation strategy is which the authors employ, even though they have a dedicated section on it (Section 2.2). They talk about injecting a "noise", without giving a definition of the said noise, nor examples. My best guess it is probably some sort of synonym-based perturbation, based on the phrase "may be close to a synonym" in Section 2.3.
- The authors claim (in Section 4.2) that their method gains SOTA on OOD datasets compared to previous approaches (Table 2). However, previous approaches are only compared with respect to in-distribution tasks (Table 1). Hence, their claim is not backed by empirical results.
- Finally, it is nowhere mentioned in the paper how many steps of training the authors perform on their mask tuning approach. Since the authors perform perturbation, their model is actually similar to a data augmentation strategy (in the related work they argue it isn't) and the model trumps the baseline due to more amount of training updates. The paper should include a discussion on this, and compare with baselines having the same amount of training steps.

## Questions

- P2, L6: "some of these methods added a complex procedure to the learning process": what is this exactly? How are those methods less complex than the one proposed by the authors?
- In Section 6.2, the authors claim their method is different from data augmentation as they do not use additional examples. However, they perturb the same training examples - isn't that also a kind of data augmentation?



**Summary Of The Paper:**

The paper proposes a fine-tuning technique which improves both in-distribution and out-of-distribution generalization of large language models on various downstream tasks. Specifically, the proposed method involves a pre-training style training during finetuning, along with the original finetuning objectives. Their proposed method appears to work well compared with the baseline on BERT, RoBERTa and BART model variants, specifically improving the out-of-distribution results.

**Summary Of The Review:**

The paper is clearly rushed and would require major rewrite to be considered for publication in this conference. While the proposed method is empirically better than the baselines, the choice of the said baselines is poor and does not reflect the central claim on out-of-domain generalization. Finally, the proposed approach is not exactly novel. While this isn't a hard negative, the lack of thorough comparison of the proposed method with similar approaches is what makes the paper a poor contribution.

---

### Official Review · Reviewer_U4CM · 2022-10-31

**Confidence:** 3
**Correctness:** 2
**Technical Novelty And Significance:** 2
**Empirical Novelty And Significance:** 2
**Recommendation:** 5

**Clarity, Quality, Novelty And Reproducibility:**

The paper is overall well-written and the research topic is valuable. The motivation is quite unclear and experiments should be improved. It is not easy to produce the results since it seems like the training process is quite unclear and there is no source code to be submitted. The reproducibility can not be guaranteed.

**Strength And Weaknesses:**

Strength:

-- The paper is easy to follow and well-writing overall.

-- The research topic on improving the generalization of PLMs is quite interesting and valuable. The proposed mask-tuning method is simple and intuitive.

Weaknesses:

-- Unclear motivation: The motivation for mask-tuning on improving the generalization is quite unclear to me. It claims that "the mask-tuning learns that training examples with different linguistic relations can have the same semantics and belong to the same label". The masked language model training loss can be large during the training in the beginning. In this case, the generated perturbed example can be high noise that doesn't semantic correlate with the original data sample. This seems like introduce the wrong perturbed examples to train the fine-tuning loss. Did the authors try to solve this or am I having any misunderstanding?  More illustration and analysis on the training process of masked language loss and fine-tuning loss in experiments are expected to help understand the mechanism of mask-tuning.

-- Unconvincing experiments: Why not use the full dev sets of the the GLUE benchmark as in the previous work HiddenCut (Chen et al., 2021)? More results on the benchmark dataset are expected to illustrate the advantage of the proposed method. Moreover, it is expected to have visualization or case studies such as the generated perturb examples to help understand the mechanism of mask tuning.




**Summary Of The Paper:**

This paper proposes a new mask-tuning method to improve the generalization of pre-trained language models, that incorporates the masked language model training with the fine-tuning. Extensive experiments on several in-distribution and out-distribution datasets with three pre-trained language models BERT, RoBERTa, and BART illustrate that mask-tuning outperforms previous methods.

**Summary Of The Review:**

An easy-to-follow paper solving an interesting problem, but with unclear motivation and unconvincing experiments.

---

### Official Review · Reviewer_Sc8b · 2022-11-01

**Confidence:** 3
**Correctness:** 1
**Technical Novelty And Significance:** 2
**Empirical Novelty And Significance:** Not applicable
**Recommendation:** 3

**Clarity, Quality, Novelty And Reproducibility:**

- There is no strong correlation between the MLM loss and the labeling consistency of original/new sentences. For example, for a sentence masking 15% of words, if only one word is predicted with its opposite word (e.g., great -> bad), the MLM loss is still low, but the label of the updated sentence would change completely.
- Why does the evaluation perform on the dev sets?


**Strength And Weaknesses:**

Strength:
- Out-of-distribution generalization is an important topic.

Weakness:
- The writing in this paper should be improved. Some explanations/descriptions are unclear.
- The major concern is reusing the label even though there is no guarantee that the predicted words would not change the label.

**Summary Of The Paper:**

This paper presents a multi-task learning method to fine-tune pre-trained LMs (e.g., BERT, BART) and evaluate on in-distribution and out-of-distribution data. In addition to the MLM loss, the proposed training process also applies a mask LM to replace randomly sampled words in an input sentence and uses the sentence's original label for standard fine-tuning with a cross-entropy loss.

**Summary Of The Review:**

The proposed methodology in this paper does not seem correct to me, as there is no guarantee that the predicted words would not change the label. Fine-tuning with the proposed method may reinforce the errors made by MLM. Even though the paper shows better results of their method on the selected datasets, it's questionable whether it could be reliably applied to other datasets.

---

### Official Review · Reviewer_ahpL · 2022-11-02

**Confidence:** 4
**Correctness:** 2
**Technical Novelty And Significance:** 2
**Empirical Novelty And Significance:** 2
**Recommendation:** 3

**Clarity, Quality, Novelty And Reproducibility:**

The paper is not well written and the experiments are of low quality. The work also seems just marginally novel (joint training of MLM + finetuning objective instead of sequential - the sequential approach is fairly standard. Unfortunately , there's no comparison to that too). There's poor justification of the design choices, baselines chosen, benchmarks selected, and how the results support the claim.

**Strength And Weaknesses:**

Strengths:
* The idea proposed in this work is fairly simple, consequently simple to implement and test widely.
* Decent ablations are present for the proposed idea and the method achieves good performance on the OOD datasets.
* The experiment details are adequately present with different hyperparameters.

Weaknesses:
TLDR:
* This paper is not well written. It was hard to follow, some sections are not clear and hard to understand, and there is significant abuse of standard terminology throughout the paper. There are also significant grammatical and language issues (I have put them separately and won't consider them as weakness if they can be fully addressed). This paper needs significant polishing.
* The claims made in this paper are not well justified (see details below). The results are weak and don't justify the main claims.
* The approach does not seem to be very novel: Use MLM to perturb examples, Jointly train MLM and Finetuning (but I will give the benefit of doubt to the authors as I'm not aware of the related work extensively).
* The authors claim that a low MLM loss signifies that the model predicts a token semantically close to the original token. That is not true. The MLM loss does not take semantic similarity of words into account. (see 2.3 2nd para line 1, 2.1 last para 3rd last line).
* They hypothesize that "MLM creates realistic and diverse perturbed examples that considerable modify the relation between the input token without affecting the label" (Section 2.2)-> but have not shown any examples of it. Nor is any modification of any relation happening.
* The choice of datasets raises questions. It is claimed that "OOD datasets are drawn from a different distribution with more linguistic variation than in distribution sets" (line 4 section 3.1). Q) How do you measure this linguistic variation? Is the OOD datasets chosen standard?
* Baseline models have not been explained at all. Is the baseline that "does pretraining using MLM on the target dataset followed by standard finetuning" included? Are the baselines models, or just Data augmentation techinques that can be plugged on top of other models?
  * Hard to compare your results with the baselines as most of the numbers are not there for OOD dataset. It is claimed that the numbers are missing because the baselines have not reported them, but to fully justify the efficacy of this method, i urge the authors to implement / run the baselines on the given datasets for a decent comparison. For instance, if any baseline is just doing data augmentation using some technique, I feel it can very well be run on top of BART to get the results of that "data augmentation technique" for comparison. All in all, I'm not convinced by the quality of comparison to baselines. Why is there no "transfer learning" baseline (or the one I suggested - MLM to improve representation learning and then FT)?


Details and questions:
* What does "linguistic relations" even mean? It is claimed multiple times that "Mask tuning perturbs the linguistic relation of the examples in the downstream task". I was not sure what this even meant until probably after the experiments section when I realized that just some tokens are changed. I don't think changing some tokens is related to "linguistic relations" or just perturbing tokens can be claimed as "perturb the linguistics relation between input examples".
* The word semantics is used very lightly. To be precise, the authors claim that "training examples have same semantics if they belong to the same label". That is absolutely not true. Semantics of text does not mean that they predicting the same label.  (mentioned at: last line of intro before contributions, Section 2.3 line 4)
* What does "biased keyword" in last line of 1st para of intro mean? This is not explained at all. I understood this at the very end after reading the related works!
* Intro second para : "c)  They (baselines) need to re-run the pretraining phase from scratch". Why? I think this is just the "finetuning stage". This is not adequately justified.
* 3rd para of intro line 2 - "joint-training improves" - Q) joint training of what?
* Intro para 4: Line 3 - "MLM loss to incentivize the FT training loss when" - Q) incentivize to do what?
* Section 2.1 line 3 - "perturbs the relation between the input tokens using the" -Q) what does this even mean? Also, you're never perturbing the relations between tokens. You're just perturbing the tokens.
* Section 2.1 para 3 - p(y'=y|x) is not the "fine-tuning function".
* What does "for passing each training iteration" even mean? (eg: usage in first line of page 4, section 2.3).
* Section 2.2 -hypothesis that MLM modify the relation between input token by generating realistic and diverse perturbations is not verified at all. There are no qualitative examples of pertubations. No stats on how many tokens are *actually* perturbed by the LMs. etc.
* In 2.3, 2nd para : "Mask tuning loss is large enough to prompt masked modeling to generate further variations and fine-tuning model to train further to handle variations" - It was never discussed that those examples without variations are dropped. Or any kind of "further training" on some specific examples is performed. Q) Are you filtering examples at all?
* In 2.3 3rd para line 1 - again wrong assumption. High MLM score does not mean predicted tokens are not semantically related.
* Q) why did you evaluate on dev set and not test sets? Do the baselines also report results on the dev set?
* Explain why the chosen datasets are OOD. Eg: Are they created from different domains? (news vs social media)?
* The OOD Phrase identification dataset "has a high lexical overlap but different semantic meanings" Q) How was this judged? Does this really make the dataset OOD? (I doubt).
* In 3.2 line 5 : "larger model does not necessarily reduce the generalization gap. Thus we choose base" - How is this a justification to use the base model?
* What is the Original finetuning model? It is just specified in 3.2 " We used Wolf et al. (2020) for the original finetuning" Please describe this approach in a sentence.
* In Related work: Please talk a bit about how your work is different from domain adaptation.
* The claim in related work Data augmentation 6.2 last line : "MLM does not use additional training examples" -> I'm pretty sure that there are data augmentation works that also dont add new examples, but just perturb existing ones (but I might be wrong). I urge the authors to double check this.

Grammar and language:
*Abstract:
  * Abstract line 1: "Pretrained language models have the known generalization problem" : Q) to what? Domain, languages, tasks, etc? It is later clarified (OOD datasets) but good to add it in this sentence.
* Intro:
  * Line 5 of intro - "pretrained language models trained on the specific dataset" - Q) pretrained or finetuned?
  * Lines 6-8 in intro are confusing.
  * Line 9 in intro - "solutions that have ~been~ referred the generalization gap as" - This sentence does not make sense.
  * last 4 lines of para 3 are hard to read. Not sure what is meant in the last line ("As a result, mask tuning is applicable with any pretrained lm that works with the original fine-tuning").
  * Para 4 line 4 - "correct output likely be according to" -> "correct output is likely due to"
* 2.1
  * Para 3 line 1 - "called perturbed example, fed into fine-tuning to be classified based on true label y" - this sounds weird.
  * last line in page 3 - "<equation>\nWhere \alpha is a weighting factor, we employ it" -> "<equation>,\nwhere \alpha is a weighting factor. We employ it ..."
  * on the one hand -> on one hand.
  * last line - "completed wrong predicted tokens" -> "wrongly predicted tokens".
* 2.2
  * line 6 - "successful masked language modeling that" -> insert "objective" between modeling and that.
* 2.3
  * Be consistent with your usage of your own terminology. Sometimes "fine tuning loss", sometimes " fine loss"  is used (section 2.3).
* 3.2
  * Line 2 of 2nd para - "fine tuning step, learning rate" -> "fine tuning step where learning rate".
  * 5th last line character 1 - "o.6" -> "0.6"
* 4.2
  * last line : "but" ->", sometimes, "


Other suggestions:
* Make references to figure, table as links.
* The "mask tuning" nomenclature is confusing in Section 5 (2nd last para) as it refers to the main proposed approach as well as one component of the loss of that approach.


**Summary Of The Paper:**

This work proposes an approach called "Mast Tuning" that combines the objectives of masked-language modeling (MLM) and standard fine-tuning (FT) to train a more robust model that generalizes to out-of-domain (OOD) datasets well. They take a pretrained language model, perturb the training data of the finetuning stage and jointly train on the MLM and FT objectives. The finetuning is performed on the perturbed examples with the hypothesis that after the masked-LM prediction stage, they are sufficiently diverse. The method outperforms 8 existing approaches on 3 downstream tasks (sentiment analysis, paraphrase detection, NLI) on a total of 5 out-of-domain datasets.

**Summary Of The Review:**

The paper needs significant clarifications and rewriting to make it more clear and easy to read. There are many crucial issues and the claims made in this paper are not well justified by the experimental choices. Many unfound claims are made in this work. However, the idea seems very simple and should be thoroughly investigated because it can have a strong impact on training general models. I urge the authors to take my review in a constructive manner to improve this work.

---

### Decision · Program_Chairs · 2023-01-20

**Decision:**

Reject

**Justification For Why Not Higher Score:**

Major issues regarding the execution

**Justification For Why Not Lower Score:**

N/A

**Metareview: Summary, Strengths And Weaknesses:**

The paper proposes a new mask-tuning method to improve the generalization of pre-trained language models, that incorporates the masked language model training with the fine-tuning. Reviewers raised several issues regarding clarity, experiments, and soundness, which suggests that the paper is not ready for publication yet and would benefit from future iterations.